# Applications of Solution NMR in Drug Discovery

**DOI:** 10.3390/molecules26030576

**Published:** 2021-01-22

**Authors:** Li Shi, Naixia Zhang

**Affiliations:** 1Department of Analytical Chemistry, Shanghai Institute of Materia and Medica, Chinese Academy of Sciences, Shanghai 201203, China; shili@simm.ac.cn; 2University of the Chinese Academy of Sciences, 19A Yuquan Road, Beijing 100049, China

**Keywords:** nuclear magnetic resonance (NMR), fragment-based drug discovery (FBDD), ligand-observed NMR, target-observed NMR, working mechanism elucidation

## Abstract

During the past decades, solution nuclear magnetic resonance (NMR) spectroscopy has demonstrated itself as a promising tool in drug discovery. Especially, fragment-based drug discovery (FBDD) has benefited a lot from the NMR development. Multiple candidate compounds and FDA-approved drugs derived from FBDD have been developed with the assistance of NMR techniques. NMR has broad applications in different stages of the FBDD process, which includes fragment library construction, hit generation and validation, hit-to-lead optimization and working mechanism elucidation, etc. In this manuscript, we reviewed the current progresses of NMR applications in fragment-based drug discovery, which were illustrated by multiple reported cases. Moreover, the NMR applications in protein-protein interaction (PPI) modulators development and the progress of in-cell NMR for drug discovery were also briefly summarized.

## 1. Introduction

Nuclear magnetic resonance (NMR) spectroscopy has been widely used in structure determination and dynamics investigation of biomacromolecules under physiological conditions. Meanwhile, due to its advantages in detecting transient and weak interactions, NMR has been becoming a powerful tool in drug discovery [1,2,3,4]. FBDD (fragment-based drug discovery), which serves as a key approach for finding high-quality lead candidates, has benefited a lot from NMR spectroscopy development [5,6,7,8,9,10]. Accumulated studies have shown the extensive applications of solution NMR in FBDD field (Figure 1), which include fragment library construction, ligand-observed and target-observed hit screening and validation, etc. [2,11]. During the past decade, FBDD has established itself as a promising drug discovery approach, which has been applied in candidate compound developing for various drug targets such as DNA, RNA, kinases, enzymes, membrane proteins, and even inherently disordered proteins [12,13,14,15]. Fragment compounds for FBDD are small organic molecules with their molecular weights typically not exceeding 300 Da, and due to the limited molecular size of fragment compounds, their binding affinities to the targets usually fall into the micromolar to millimolar range [16]. NMR spectroscopy, which is sensitive to weak interactions, is one of the top choices for hit compound screening against a fragment compound library [17]. Meanwhile, since target-observed NMR techniques are capable of providing structural information for structure-guided hit fragment optimization [18,19], they are alternative methods to X-ray crystallography for the characterization of target-hit/lead interaction. Dynamics investigation, inter-molecular NOEs (nuclear Overhauser effects) and paramagnetic NMR can help to reveal atomic level details associated with the binding mode of the hit or lead to defined target [20,21,22,23,24]. Although NMR data collection and data processing is time-consuming, this is especially necessary when researchers devote great efforts to obtain the crystal structures of target-hit/lead complexes but in vain [25].

Except for the conventional target-oriented drug discovery, targeting protein-protein interactions (PPIs) has been emerging as an attractive approach for drug development. The protein-protein interaction network, the so-called interactome, participates in extensive biological processes, and aberrant expression or regulation of the interactome would cause the occurrence of severe human diseases [26,27,28,29]. PPI modulators, which present increased target or signal pathway selectivity and decreased off-target side effects, have high potentials for therapeutic uses [30]. Hence, the discovery and development of chemical compounds to modulate the interactome has gained enormous attention. Compared with the typical binding pockets for small molecules in conventional targets, the surface areas for protein-protein interactions are often large and flat, which introduces more challenges to the PPI-targeted drug development [31,32,33]. However, it is worth mentioning that NMR and FBDD have been becoming powerful tools in developing drugs for the ‘’undruggable’’ targets due to their advantages in dynamic and transient systems such as protein-protein interactome [26,34,35,36]. Multiple PPI-targeted hit compounds against fragment compound libraries have been developed [37,38,39,40,41], and a few of PPI stabilizers and breakers have been validated by NMR [38,39,42]. GNE667, a novel inhibitor of deubiquitinase USP7, was found to disrupt the interaction between USP7 and its native substrate ubiquitin [38]. CC0651 inhibited the activity of Cdc34A (an E2 enzyme) by enhancing the binding affinity of ubiquitin to Cdc34A, thus blocking the discharging process of ubiquitin to E3 ligase [39].

Similar to the *in vitro* NMR applications in drug development, NMR techniques could also be conducted to detect target-compound interactions in living cells such as *E.coli*, yeast and mammalian cells [43,44]. Researchers have applied in-cell NMR techniques in compound screening and target engagement, which are two important aspects of drug discovery [43,45]. However, there are still several factors need to be considered as challenges remaining in the in-cell NMR studies, which include molecular size limitation, high background signals, nonspecific interactions in cells and protein leakage produced by dead cells [44,46,47,48,49].

In this manuscript, we reviewed the current progresses of NMR applications in fragment-based drug discovery, which include the NMR techniques utilized in fragment compound library generation, the NMR methods for hit screening and validation, and the NMR experiments applied in hit optimization and active compound working mechanism elucidation (Figure 1). In addition, the NMR applications in PPI modulators development and the progress of in-cell NMR for drug discovery were also briefly summarized.

## 2. NMR in Fragment-Based Drug Discovery

Over the past two decades, FBDD has emerged as an efficient approach to discover high-quality leads for drug development. Currently, a total number of four drugs developed using the FBDD method have been approved by the U.S. Food and Drug Administration for clinical use, which are Vemurafenib (approved in 2011), Venetoclax (approved in 2016), Pexidartinib and Erdafitinib (both approved in 2019) [7,50,51]. Besides, tens of drug candidates derived from FBDD have been advanced to clinical trials so far [10,51]. Compared with the conventional high throughput screening method, FBDD tries to identify small, weakly binding fragment compounds with a high ligand efficiency, and these fragment hits can be optimized efficiently into potent leads by linking, merging and growing [52,53]. FBDD has several advantages over traditional high throughput screening approach: (1) the small molecule library gives access to explore broader chemical space; (2) high hit rate; (3) high ligand efficiency; (4) a better chance to optimize the small molecule to have a drug-likeness parameters [54,55,56].

### 2.1. Fragment Compound Libray Construction and Group Generation

As it has been well known, most of the drugs present drug-like properties normally defined as the Lipinski’s rule of five (RO5). According to the experimental practice during the past two decades, the fragment compounds admitted to the fragment library are suggested to follow the criteria of the rule of three (RO3): (1) molecular weight ≤ 300 Da; (2) clogP ≤ 3; (3) hydrogen bond donors and acceptors each ≤ 3; (4) the number of rotational bond ≤ 3 [16,51]. Fragment compounds can be designed to fulfill specific research interests or purchased from vendors, such as Enamine, Maybridge, and ChemBridge. The capacity of the fragment library can vary from hundreds to thousands of molecules according to demands, but the cost-effective library size is about 2000 if regular fragments are selected [57,58]. Currently, many pharmaceutical companies and research laboratories have generated their own fragment libraries to establish the FBDD platforms. Multiple types of fragment libraries, which include natural product fragment library, protein-protein interaction library, fluorinated library, brominated fragment library, etc. were setup [41,59,60]. Although there are no strict constraints for the chemical structures of fragments incorporated into the library, some pan-assay interference compounds (PAINs), such as nonspecific binders, reactive covalent modifiers, chelators, and aggregators, should be avoided [61]. Sometimes, it is easy to identify PAINs, such as those compounds containing a Michael acceptor, alkyl halide, or epoxide, which are chemically reactive to electron donors [10]. However, in most circumstances, the promiscuous behavior is non-obvious. Therefore, a La assay to detect reactive molecules by nuclear magnetic resonance (ALARM NMR) and computational filters have been utilized to remove PAINs compounds from libraries [62,63,64].

In practice, many biophysical techniques have been used to conduct the first round of screening, which include surface plasmon resonance (SPR), thermal shift, weak affinity chromatography - mass spectroscopy (WAC-MS), X-ray crystallography, microscale thermophoresis (MST), and NMR, etc. [65,66,67,68,69]. The advantages and drawbacks of these techniques are briefly summarized in Table 1. Compared to other techniques, target-based NMR and X-ray crystallography tend to have low false positive rates. If NMR is used as the primary screening method, extra rules need to be followed. For conventional proton-based NMR screening, multiple fragments with no significant proton signal overlapping are pooled into one group to improve the screening efficiency. In our lab, to generate our own fragment library, all of the small compounds in the ZINC database were filtered according to the modified RO3 [70]. Then, to cover a broader chemical space, the resulting fragments were clustered into groups according to their structural similarities, and only those cluster-center compounds were selected and purchased for fragment library construction [70]. Finally, after NMR evaluation, the fragment compounds with their water solubility larger than 100 μmol/L in phosphate buffer (20 mmol/L NaH_2_PO_4_/Na_2_HPO_4_, 100 mmol/L NaCl, 2% dimethyl sulfoxide (DMSO), pH 7.4) were clustered into 56 groups (8–10 compounds in each group) by following the rule of no significant NMR signal overlapping in the spectra of mixed group compounds [70]. The grouping of fragment compounds could also be done using the assistance of a computer. Xavier Arroyo et al. and Jaime Stark et al. presented a fast and straightforward computer-aided method to design and optimize the mixtures of fragments with minimized NMR signals overlapping [71,72]. Similar to proton, Fluorine-19 is also a 1/2 spin nucleus. ^19^F is the second most sensitive stable NMR-active nucleus with a high abundance (100%) in nature. As the fluorine atom is quite rare in biomacromolecules and less signal overlapping is expected in the ^19^F spectrum, the ^19^F probe has caught researchers’ eyes [73,74,75]. Interestingly, there is generally no strict pooling strategy for fluorine-based fragments, and the number could be safely extended to more than 20 per group [76,77]. After fragment library construction, it is important to take the quality control into accounts for maintaining the quality of the compound library [78]. Normally, the compounds dissolved in DMSO-*d*_6_ are suggested to be stored in a freezer (−20 °C), and the frequency for freeze-thaw cycles should better be controlled. 

### 2.2. Ligand-Observed or Target-Observed Hit Generation and Validation

Ligand-observed NMR and target-observed NMR are two classes of approaches widely used in fragment hit screening and fragment hit validation [6,83]. Ligand-observed NMR spectroscopy detects the NMR behavior changes of ligand compounds upon the presence of target biomacromolecule. Currently, multiple ligand-observed NMR techniques including Carr-Purcell-Meiboom-Gill pulse sequence (CPMG), saturation transfer difference (STD) and water-ligand observed via gradient spectroscopy (WaterLOGSY), etc. have been applied in fragment hit screening and validation [37,84,85]. 

The Carr-Purcell-Meiboom-Gill experiment (CPMG) is a relaxation-edited, ligand-observed NMR technique that employs differences in relaxation properties between the nuclei in biomacromolecules and the nuclei in small compounds to probe binding [86]. Small compounds tend to relax slowly and exhibit sharp, well-defined peaks, whereas biomacromolecules and their bound ligands usually relax rapidly. Therefore, a CPMG pulse sequence is applied to filter out the signals of target biomacromolecule and its bound ligands without significantly affecting the signals of unbound small molecules. Hence, this approach can be conducted to detect the signal intensity decreasing and resonance shifting of hit compound upon its binding to the target biomacromolecule (Figure 2a) [86,87,88]. CPMG experiments work well when the compound is in 10 to 20-fold molar excess of the target. This ratio is important because a larger compound-target ratio may mask binding due to the strong signal from the free compound [86,89]. As that shown in Figure 2a, active compound SOMCL-16-171 exhibited substantial line broadening and signal shifting in the recorded CPMG spectrum upon its binding to Hsp82, which is a yeast homologue of human Hsp90 [90]. Saturation transfer difference (STD) spectroscopy is another commonly used ligand-observed NMR method in the first round of fragment hit screening and validation [85,91]. It detects the inter-molecular magnetization transfer by comparing the difference of two recorded spectra of ligand compounds with and without the saturation of target protein signals. Generally, the protein-specific protons at upfield (typically around 0 ppm to avoid small molecule resonances) are saturated by selective irradiation, and the resonance saturation will then rapidly spread over the entire protein, ultimately leading to transferred NOEs with protons of the ligand compound binding to target protein (Figure 2b). The STD experiment is sensitive to off-rate kinetic constant of the binding process [85]; generally, an appropriate range of the dissociation constant between the target and ligand for STD experiment is about 1 mM to 0.1 µM [85,92]. As shown in Figure 2b, fragment hit 1-E6, which is a weak binder to the middle domain of Hsp90 (Hsp90M), exhibited positive STD signals upon the presence of Hsp90M [90]. In our group, CPMG and STD were jointly applied to discover fragment hits targeting BRD4, Hsp90M (Hsp90α’s middle domain), PDEδ, USP7. Multiple fragment hits with new scaffolds against the aforementioned protein targets were identified [37,70,90]. Among four target proteins, Hsp90M (Hsp90α’s middle domain) is the so-called ‘’undruggable’’ protein. By NMR screening, one hit compound (1-E6) was identified from the fragment library containing 539 compounds, and Hsp90M-targeted allosteric modulators (SOMCL-16-171 and SOMCL-16-175) derived from 1-E6 were developed for the first time.

In addition to CPMG and STD experiments, water-ligand observed via gradient spectroscopy (WaterLOGSY) has also been applied in fragment hit screening and validation [84,93]. As the STD approach, WaterLOGSY use the “through-space” NOE to monitor ligand binding to targets. WaterLOGSY relies on the transfer of magnetization between water, the target, and the compound of interest via NOE and chemical exchange. By selectively saturated the bulk water molecules in solution, different NOE signals between small and large molecules will be exploited [93,94]. Of note, although the aforementioned ligand-observed NMR experiments have established themselves as powerful tools for hit screening and validation, they also have limitations. To improve the authenticity and avoid false positive results introduced by nonspecific binding and compound aggregation, competition and dose-dependent experiments should be considered [95,96].

In addition to the widely used proton-observed NMR, fluorine-19 NMR techniques have also demonstrated themselves as powerful tools in fluorinated fragment library screening and validation [97,98]. The fluorine atom is frequently applied in molecular design throughout the lead-optimization phase in drug discovery, and the existence of fluorine atom can critically influence pharmacokinetic and/or pharmacodynamic properties of the designed compound [99]. John Jordan et al. demonstrated that ^19^F-based NMR is not only a rapid and sensitive method in detecting fragment hits but also can provide structure-activity relationship (SAR) information for hit-to-lead optimization [76]. Fluorine chemical shift anisotropy and exchange for screening (FAXS) is a fluorine-NMR competition binding experiment that requires the use of a fluorinated spy compound to defined target. The spy compound binds to the target with a medium to low affinity. When the competitive ligand binds to the target protein, the displacement of the fluorinated spy compound will occur. Therefore, intensity recovery of the NMR signal from the spy compound will be an indication of specific binding [76]. By using the FAXS technique, Elena Casale et al. developed novel HSP90 inhibitors [100]. N-Fluorine atoms for biochemical screening (n-FABS) is also a highly sensitive NMR technique that has been used for fragment screening and compound inhibition activity determination. N-FABS is a biochemical method that requires the labeling of the enzyme substrate with a fluorine-containing group [101]. Chiara Lambruschini et al. has utilized this method to discover inhibitors targeting the membrane enzyme fatty acid amide hydrolase (FAAH) [102]. 

Target-observed NMR spectra provide the chemical shift information of ^1^H and ^15^N atoms in the backbone amide groups of the target protein. The ^1^H-^15^N hetero-nuclear single quantum coherence spectrum (HSQC) and ^1^H-^15^N hetero-nuclear multiple quantum spectrum (HMQC) serve as fingerprints of target proteins [103]. Compared to ligand-observed NMR, target-observed NMR has some disadvantages for the first round screening purpose. (1) The protein needs to be isotope labeled, which increases the cost of screening. (2) Compared with that of small molecules, the relaxation rate of macromolecules is much faster. Line broadening and signal overlapping will become a substantial obstacle in NMR data collection when the molecular weight of the target macromolecule exceeds the threshold value. Up to date, molecular weight limitation remains a question for data collection though multiple isotope labeling strategies, and NMR techniques have been jointly used. (3) Screening by using target-observed NMR is quite time consuming [104]. However, the target-observed NMR approach is a good choice when a known binding site has already been elucidated or to study the “structure and activity relationship (SAR by NMR)” of a given target [4,105]. Selena Simon et al. collected the HMQC spectra of ^15^N-labeled WDR5 in the presence of 12 fragment mixtures. After a paralleled comparison to its interaction with MYC, several groups were deconvoluted by using a time-consuming single compound validation approach, and hit fragments were selected as the starting points to design WDR5-MYC PPI inhibitors [106]. The pioneering work of Stephen Fesik et al. in 1996 utilized the ^1^H-^15^N HSQC approach to develop high-affinity ligands binding to FKBP (FK506- and rapamycin-binding protein) [4], and this paper also introduced the phrase “SAR-by-NMR” to the field for the first time. With the guidance of binding site information extracted from ^1^H-^15^N HSQC analysis, compounds with nanomolar affinities for FKBP were rapidly discovered by tethering two ligands with micromolar affinities. 

As previously described, the ^1^H-^15^N hetero-nuclear single quantum coherence spectrum (HSQC) and ^1^H-^15^N hetero-nuclear multiple quantum spectrum (HMQC) serve as fingerprints of target proteins [103]. The chemical shifts of backbone amides are sensitive to chemical microenvironment changes. When a binder is mixed with the isotope-labeled target protein, chemical shift perturbations for amide resonances will be induced by ligand binding or ligand-induced conformational changes [107]. If the backbone assignments and the structure of target protein are known, target-observed NMR data can provide the binding sites information of a specific ligand in the target protein [108,109]. In addition to its application in binding site recognition, target-observed NMR could also be used to determine the binding affinity of ligand to target protein. Ligand binding is a dynamic and reversible process, and the on-rate (association) and off-rate (dissociation) are intimately coupled to the binding potency of ligand to target. According to the different binding potencies, three types of chemical shift perturbations (CSPs) will be observed (Figure 3a). For the weak binder (Kd in high micromolar to millimolar range), a fast exchange between a target in the apo state and a target in the ligand-bound state is expected, and the gradually shifted chemical shifts for the perturbed residues in the target protein will be observed with the addition of increased amounts of weak binder. The dissociation constant Kd for the weak binder:target protein system could be determined according to Equation (1) [110]. Here, [P] is the total protein concentration, [L] is the total ligand concentration, CSPmax is the maxium CSP observed for a saturated state, and CSPobs is the observed CSP at a particular ligand concentration.
(1)$${\mathrm{CSP}}_{\mathrm{obs}}{/\mathrm{CSP}}_{\max}\;=\;(\mathrm{Kd}\;+\;[L]\;+\;[P]\;-\;\sqrt{{\left(\mathrm{Kd}\;+\;\left[L\right]\;+\;\left[P\right]\right)}^{2}\;-\;4\left[L\right]\cdot \left[P\right]})/2[P]$$

As shown in Figure 3b, we titrated increased amounts of SOMCL-16-175 into isotope-labeled Hsp90M and recorded the HSQC spectrum for each titration sample. Then, the chemical shift perturbation values of several residues in Hsp90M that exhibited substantial chemical shift changes were applied to calculate the binding affinity of SOMCL-16-175 to Hsp90M according to Equation (1) [110]. A zoomed-in view of the titration spectra for residue L328 in Hsp90M was presented for illustration purpose, and the binding affinity of SOMCL-16-175 to Hsp90M was determined (Kd = 804 ± 24 µM) accordingly [90]. For the binder with a moderate binding affinity (Kd in micromolar range) to the target protein, an intermediate exchange between the target in the apo state and the target in the ligand bound state is expected, and the gradually shifted and attenuated chemical shifts for the perturbed residues in the target protein will be observed with the addition of increased amounts of binder. While for the strong binder (Kd in the low micromolar to nanomolar range), a slow exchange between the target in the apo state and the target in ligand-bound state is expected, and the attenuated signals corresponding to the apo state and the appearance of new signals corresponding to the ligand-bound state will be detected. Although the binding affinities of moderate and strong binders to the target protein could not be accurately determined by using a traditional target-observed NMR method, a rough ranking of the binding potencies of different hit compounds to the target protein could be estimated according to the recorded spectra [103]. Of note, in the year of 2020, Gary Pielak’s group successfully quantified the kinetics and equilibrium thermodynamics for the binding of a fluorine-labeled Src homology 3 (SH3) protein domain to four proline-rich peptides by doing one-dimensional ^19^F NMR lineshape analysis [111]. This approach is capable of determining the binding affinities of moderate binders to the target protein and may serve as a valuable tool in the NMR drug discovery and development.

Compared with ^1^H-^15^N HSQC spectroscopy, which is the most frequently used target-observed NMR technique in drug discovery, ^1^H-^13^C HSQC spectroscopy can also serve as an alternative technology in target-observed NMR screening, although the substantial cost of ^13^C labeling by using ^13^C-glucose as the carbon source typically limits its application in FBDD. Stephen Fesik reported a cost-effective [^1^H, ^13^C] NMR screening strategy that significantly increase the sensitivity by nearly 3-fold compared with that of NMR-based screening using [^1^H, ^15^N] HSQC. This method involves the selective ^13^C labeling of methyl groups in valine, leucine and isoleucine residues of target protein by introducing [3,3,-^13^C]-α-ketoisovalerate and [3-^13^C]-α-ketobutyrate as amino acid precursors which are biosynthetically incorporated into protein when it is expressed in bacterial systems [112]. Besides ^1^H-^15^N HSQC and ^1^H-^13^C HSQC approaches, protein-observed ^19^F NMR (PrOF) has also been applied in fragment screening and validation [113]. Fluorine can be incorporated into target protein through sequence-selective replacement of the natural amino acid with a fluorinated variant or via conjugation of fluorine-containing small molecules to protein side chains [114]. Neeraj Mishra et al. introduced fluorinated aromatic amino acids (5-fluorotryptophan or 3-fluorotyrosine) into the bromodomain of BRD4, and validated the interactions between JQ1 and 3FY-BRD4 or 5FW-BRD4 [115].

### 2.3. Hit-to-Lead Optimization

In FBDD study, multiple hits could be obtained after the first round screening and the second round cross-validation by using NMR and other biophysical methods. For a better evaluation of the fragment hits, some metrics have been devised taking compound parameters into account. In addition to the biological activity of the hit, ligand efficiency (LE) is another important consideration. Ligand efficiency is defined as the binding energy divided by the number of non-hydrogen atoms in the molecule. Therefore, ligand efficiency considers both the potency and the size of the molecule. Hit compounds with higher LE values indicate more potential improvement in binding affinity and higher possibility to achieve drug-like properties during hit-to-lead optimization [116]. Usually, LE ≥ 0.3 is considered as a suitable starting point for hit-to-lead optimization [117,118]. Due to the low binding affinity of fragment hit to target, it should be optimized with the assistance of the medicinal chemistry approach [119]. Generally, there are three major strategies to improve the binding affinity of hit compounds to the target, namely growing, merging, and linking, as shown in Figure 4 [118]. For the growing strategy, it allows extending the molecule around a single hit. For the linking strategy, it needs to find two hits that bind to different but adjacent pockets in the target. Then, different types and variant lengths of linkers are chosen to link two hits together. It is capable of substantially increasing the binding affinity according to Gibbs function when the appropriate linker is used [120]. For the merging strategy, it combines the common structure parts of overlapping hits. In practice, these three strategies can be jointly applied to generate new chemicals. To improve the efficiency of the hit-to-lead optimization process, the structural information for the hit:target complex should better be provided. X-ray crystallography and the alternative NMR approach are two major tools to achieve structural information in FBDD. For NMR, ^1^H-^15^N, and ^1^H-^13^C hetero-nuclear correlation spectra such as [^1^H, ^15^N]-HSQC, [^1^H, ^15^N]-TROSY-HSQC, and [^1^H, ^13^C]-HSQC etc. are widely used experiments to provide binding sites information [121]. By performing chemical shift perturbation analysis and mapping residues that undergo substantial CSP changes onto the target’s structure, the binding sites of the fragment hit to target could be elucidated. In addition to mapping the binding sites, the structure of the hit fragment complexed with the target protein could be solved by a combination use of 2D and 3D NMR experiments. However, since solving the complex structure by NMR is not only time consuming but also limited by some other disadvantages, X-ray crystallography is usually the top choice to determine complex structures [21]. Serena Monaco et al. developed a novel protocol named Differential Epitope Mapping by STD (DEEP-STD) to identify the types of protein residues (aromatic, polar, nonpolar) contacting the ligand [122]. This method can be used to rapidly reveal pharmacophore information responsible for ligand binding. Whether the 3D structure of the protein is known or not, it can help to orient the ligand in the target protein [123].

Anders Friberg et al. found two distinct classes of hits that were shown to bind to two different regions of Mcl1 by target-observed NMR screening (Figure 5). The class 1 fragment hits contain 6,5-fused heterocyclic carboxylic acid. The class 2 fragment hits contain a hydrophobic aromatic system tethered by a linker to a functional group, most often a carboxylic acid. They acquired NOE-derived distance restraints and docked these two different classes of fragments onto Mcl1 by Xplor-NIH. By inspecting the NMR models, they found that class 1 and class 2 fragments bind to the proximal subsites in a large pocket of Mcl1. Then, they optimized two hit fragments (compound **2** and compound **17**) by using merging and linking strategies and produced compound 60 (Figure 5). By a further chemical elaboration on compound 60, they generated compound binds to Mcl1 with a dissociation constant of < 100 nM [124]. Andreas Frank et al. discovered two different binding sites when conducting the ^15^N-RPA70N observed HMQC screening and chemical shift perturbation analysis. By solving high-resolution ternary X-ray crystal structures, they revealed the binding modes of Hit 1 and Hit 2, which were used to guide the hit optimization [125]. However, the availability of high-resolution complex structure is sometimes still a bottleneck in drug discovery process. Morkos Henen et al. developed the meta-structure analysis of primary sequences approach and fragment-based NMR spectroscopy AFP-NOESY (Adiabatic Fast Passage pulse to probe ^1^H-^1^H NOE), which provided an alternative method for the rational design of fragment evolution without resorting to highly resolved protein complex structures [126]. Computational approaches have also been used to guide hit-to-lead optimization [127,128].

### 2.4. Working Mechanism Elucidation 

When a satisfied lead is available, to elucidate its mechanism of action at atomic details, the binding mode of lead compound to the target protein needs to be investigated. When X-ray crystallography fails in solving the complex structure of lead compound bound to the target, NMR techniques such as RDC (Residual Dipolar Coupling), PRE (Paramagnetic Relaxation Enhancement), inter-molecular NOEs, etc. could serve as alternative approaches to reveal the structural information [24,129,130,131]. RDC provides multi-domain orientation information in a nonisotropic environment [132,133]. Paramagnetic labeling reagents are molecules that contain at least one unpaired electron. The unpaired electron causes significant line broadening to nearby observed nuclei, which can be applied to analyze spatial arrangement and dynamics of the system [134]. Scientists have developed various paramagnetic labeling strategies to small molecules or biomacromolecules to obtain structural information [130,135,136,137]. 

Multiple cases for NMR applications in the structural information investigation of active compound-target systems have been reported [138,139]. By conducting competition STD and INPHARMA (Inter-Ligand NOE for Pharmacophore Mapping) NMR experiments, Martina Fruth et al. validated that ureidothiophene-2-carboxylic acid bound to RNAP (a RNA polymerase) in the same pocket as its well-known binder Myx [140]. Tomohide Saio et al. carried out a detailed characterization of the ligand-induced conformational changes of a multi-domain protein MurD. To obtain long-range conformation changes information, they utilized the paramagnetic lanthanide probe to label E260C/K262C in domain 3 of MurD. Quantitative analysis of pseudocontact shifts identified a semi-closed conformational state of MurD, which is key for understanding the mechanism of native ligand binding [141]. Nathalie Goudreau et al. solved the solution structure of a ternary complex characterized by two inhibitor molecules binding to the two zinc knuckles of the nucleocapsid (NC) protein. Compared to the available NC/oligonucleotide complex structures, they found that their novel inhibitor mimics the guanosine nucleobases found in many reported structures [142]. Jesse Calderon et al. found DMA-135, an inhibitor targeting Enterovirus 71 stem loop 2 (SL2), induces a conformational change and stabilizes the ternary complex with AUF1 (AU-rich element RNA-binding protein 1) incorporated, thus repressing translation. By calculating the NMR structures of SL2 complexed with DMA-315 using NOEs and RDCs, they revealed that DMA-315 changes the interhelical angle between the upper and lower helices of SL2 by 77 degrees [143]. In our group, we characterized the interactions between the Hsp90 middle domain (Hsp90M) and two active compounds SOMCL-16-171 and SOMCL-16-175 by conducting chemical shift perturbation and mutagenesis analysis. We found that two loops and one α-helix (F349-N360, K443-E451, and D372-G387) in Hsp90M are responsible for the recognition of SOMCL-16-171 and SOMCL-16-175. Meanwhile, the binding of SOMCL-16-171 and SOMCL-16-175 to Hsp90M allosterically modulates the structure and function of Hsp90α’s N-terminal domain, which consequently up-regulates Hsp90α’s ATPase activity [90].

## 3. NMR in PPI Modulators Discovery

Modulators targeting the protein-protein interaction network have been becoming an important field for drug discovery [29,144]. Since the protein-protein binding surface is usually large and flat, it is quite challenging to develop PPI modulators. Due to the small size and large chemical diversity of the fragment compounds, the FBDD approach presents a high efficiency advantage over conventional drug discovery methods such as high-throughput screening (HTS). By a combination of fragment library and NMR techniques, possible hotspots in the protein-protein interaction surface can be detected, and hit fragments for further optimization will be obtained [145]. In fact, quite a few PPI modulators developed by fragment-based screening, validation, and optimization with the assistance of NMR have been reported [38,106,146]. In practice, both the ligand-observed or target-observed NMR could be used in PPI-targeted hit fragment screening. When target-observed NMR approach is applied, mixed PPI proteins with one of them selectively isotope labeled could be used to carry out hit fragment screening. By comparison of the recorded NMR spectra of selectively labeled mixed PPI protein samples without and with the presence of fragment mixtures, the possible PPI interferers would be identified [38,40]. In addition to the hit screening, 2D NMR could also be used to characterize the modulation mechanisms of hits on the protein-protein interaction system. The application of the FBDD approach and NMR techniques in PPI modulator discovery is demonstrated by the successful development of the FDA-approved drug Venetoclax (ABT-199). This compound binds to Bcl-2 and inhibits its interaction with other protein partners. During the development process, target-detected NMR spectroscopy was used to screen the fragment library, and two compounds were identified that bound to the anti-apoptotic protein Bcl-xL at adjacent sites [147]. After linking the identified two compounds and a further chemical elaboration, ABT-199 (Venetoclax), which was more selective for Bcl-2 over Bcl-xL, was obtained and approved by the FDA for the treatment of certain patients with chronic lymphocytic leukemia [148]. 

The applications of NMR techniques are not only limited to the PPI modulator discovery derived from the FBDD approach. According to the published literatures, two different types of PPI modulators, which are disruptors and stabilizers, have been developed (briefly illustrated in Figure 6) [149,150]. GNE6776 is a highly selective USP7 inhibitor. Lorna Kategaya et al. labeled the catalytic domain of USP7 and collected the ^15^N-TROSY-HSQC spectrum of USP7 in free state. Then, they mixed unlabeled ubiquitin (Ub) with ^15^N-labeled USP7 and investigated the binding mode of Ub to USP7. Subsequently, a ternary mixture sample of GNE6776, ^15^N-labeled USP7, and Ub was prepared, and the ^15^N-TROSY-HSQC spectrum was acquired. According to the NMR data, upon the presence of GNE6776, the signals of specific residues (Q287 and E371) involved in the recognition of Ub shifted back to their free state positions. This result indicates that GNE6776 is a PPI disruptor and functions by disrupting the interaction between USP7 and Ub [38]. CC0651 is an inhibitor targeting E2 enzyme Cdc34A, and it was found to enhance the interaction between Cdc34A and its substrate Ub similar to a molecular glue [39]. Although the PPI disruptor and PPI stabilizer function oppositely, their binding sites are all either on or spatially close to the PPI interaction surface. Therefore, the protein-stabilizer complex structure information can be applied to guide the designing of the PPI disruptor or vice versa. Lech Milroy et al. designed a 14-3-3/Tau disruptor 3b according to the structure of 14-3-3 complexed with the stabilizer fusicoccin A and unraveled that 3b can break the binding of pTau to 14-3-3 by conducting 2D HSQC experiments [40,149].

## 4. In-Cell NMR

In-cell NMR serves as a promising approach to provide structural and dynamics data on protein-protein interaction and protein-ligand interaction systems in cellular environments [45,49,151]. Generally, in-cell NMR can be carried out in bacteria, yeast, sf9, frog oocytes, zebrafish embryo, and human cells without further sample purification, and it has become a remarkable tool for drug discovery [45,152,153,154,155,156,157,158]. Multiple cases related to the applications of in-cell NMR in drug discovery have been reported. Enrico Luchinat et al. expressed ^15^N labeled CA2 (carbonic anhydrase) in *E.coli.* and then treated the cell with two approved drugs, acetazolamide (AAZ) and methazolamide (MZA). The spectra recorded using *E.coli.* cell samples indicated that both of the two drugs bound to ^15^N-CA2 in cells. In addition, their binding modes are similar to the mode determined in vitro and are consistent with the reported complex structure of CA2-AAZ [159]. STINT-NMR (structural interactions using NMR spectroscopy) is a unique tool for drug screening against PPI targets [160,161]. Two genes or more genes with different promotors are co-transformed into *E.coli.* The protein expression encoded by the first gene is allowed to be conducted in an isotope-labeled M9 medium, and then the second protein is expressed in unlabeled conditions. Hence, the first protein is detectable by NMR, and the second protein is not observable. This method can be applied to probe in cell protein-protein interaction and is capable of screening small molecules that potentially interfere the protein-protein interactions of interest [160]. Christopher DeMott et al. expressed ^15^N labeled Pup and unlabeled Mpa protein in *E.coli*. sequentially. Then, they used the in-cell NMR method to screen against the available compound library and analyzed the screening results by singular value decomposition (SVD). They finally got three compounds that could disrupt the interaction between Pup and Mpa in living cells [162]. The flowchart of Pup and Mpa PPI modulators discovery by in-cell NMR is shown in Figure 7. 

It is well known that research done in vitro may not accurately replicate conditions that occur in living cells. In-cell NMR study provides information on druggability and target engagement at an early stage, which might minimize the off-target side effects by excluding those unsatisfied compounds from further development [159]. Enrico Luchinat et al. applied in-cell NMR to investigate the binding of nine FDA-approved drugs to the second isoform of carbonic anhydrase (CA) in human cells. Even though all of the tested drugs bound to CA in vitro, their kinetic behaviors in living cells were strikingly different to each other. The results showed that the interplay between compound, intracellular target, membrane, and cellular milieu generated complex dynamic behaviors. Some drugs were found to gradually dissociate from intracellular CA, even under the presence of free compound in the external medium [163]. Such observations could be attributed to the off-target binding in a multiple target environment. In addition to the target-observed in-cell NMR, the ligand-observed NMR can also be performed in vivo [164]. Donatella Potenza et al. had conducted STD and trNOE experiments to validate the binding of specific compounds to integrin αvβ3 in ECV304 cells [165]. 

## 5. Conclusion Remarks

Solution NMR spectroscopy is a well-established approach to elucidate the structure, interaction, and dynamics of molecules in physiological conditions, and it has become a powerful tool in drug discovery. In fact, over the past decades, NMR has been widely used in drug-related research, especially in fragment-based drug discovery. NMR has a broader application in supporting FBDD, which is capable in fragment library construction, hit fragment screening, and binding mode characterization for the guidance of structure-based optimization [6,8,18,66]. To extend the application scope of NMR in drug discovery, scientists have devoted great efforts into the field. Isotope labeling, non-uniform sampling, reduced dimensionality techniques for rapid measurements, and automated software for NMR data analysis have been developed to improve the efficiency of NMR experiments [121,166,167,168,169]. Different NMR techniques such as selective paramagnetic labeling of target or ligand, INPHARMA, etc. have also been tried in exploring the structural information of compound/target complexes [170,171,172]. 

NMR spectroscopy has also demonstrated itself as a powerful tool in PPI modulator development. Drug development targeting protein-protein interactions has long been considered as a very difficult and even impossible task. Designing peptides mimicking amino acid residues in the PPI interface is a rational starting point to design PPI modulators; however, the bioavailability and the in vivo stability of these peptides is usually very low [29]. Therefore, small molecules targeting the PPI network have caught people’s eyes [36]. Different from the druggable pockets in conventional drug targets, the protein-protein binding surface is usually flat and undruggable. Fragmentation and NMR are robust tools for developing PPI modulators. As a convincing example, NMR-based fragment hit screening and the characterization of hit-target interactions contributed significantly to the successful development of the FDA-approved drug Venetoclax (ABT-199). 

In addition to the in vitro applications, NMR spectroscopy could also be applied to support drug discovery in a cellular context. In-cell NMR has been developed for nearly 20 years. Researchers have applied the in-cell NMR method to hit compound screening and interaction characterization of the compound/target system in different types of living cells including prokaryotes cells and human cancer cells [43,173]. Of note, severe line broadening of NMR spectra caused by the crowded cellular environment restricts the application of in-cell NMR [174]. New strategies including selective isotope labeling and high-resolution magic angle spinning (HR-MAS) etc. have been developed to improve the quality of in-cell NMR spectra [139,175]. For example, due to the low abundance in biomacromolecules and the high sensitivity, ^19^F has been utilized as an important probe to investigate target-ligand interactions in living cells [176,177]. Cell death is another issue that needs to be addressed in conducting in-cell NMR. Fast pulse sequences and various bioreactors were developed to shorten in-cell NMR data collection time and improve cell viability [178,179,180]. 

NMR, X-ray, and cryo-EM are three major structural biology techniques that have been widely used in industry and academia. Compared to NMR, X-ray and cryo-EM exhibit robust advantages on large complex systems or biomolecule machineries. However, although NMR has the limitation for biomacromolecules with large molecular weights, NMR spectroscopy presents particular merits in detecting the structural information of dynamic biomacromolecule systems. In addition, NMR is one of the most promising techniques in studying target-ligand interactions in living cells, which is pretty important for drug evaluation.

## Figures and Tables

**Figure 1 molecules-26-00576-f001:**
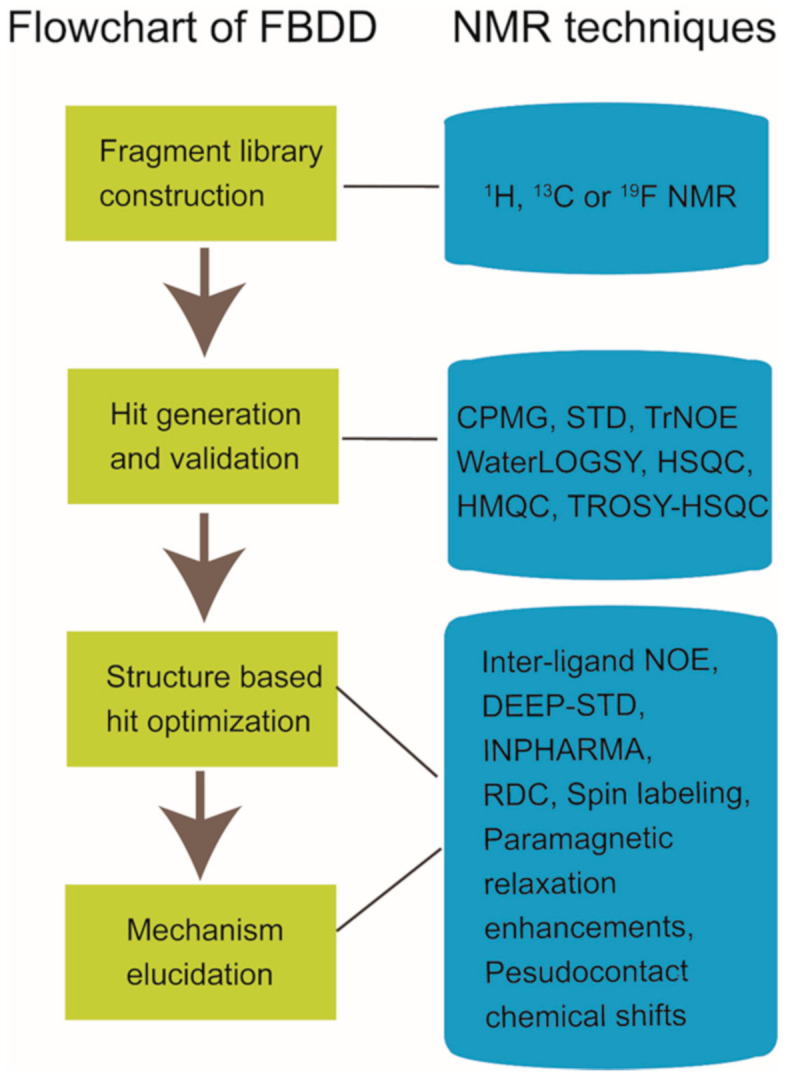
Nuclear magnetic resonance (NMR) applications in fragment-based drug discovery.

**Figure 2 molecules-26-00576-f002:**
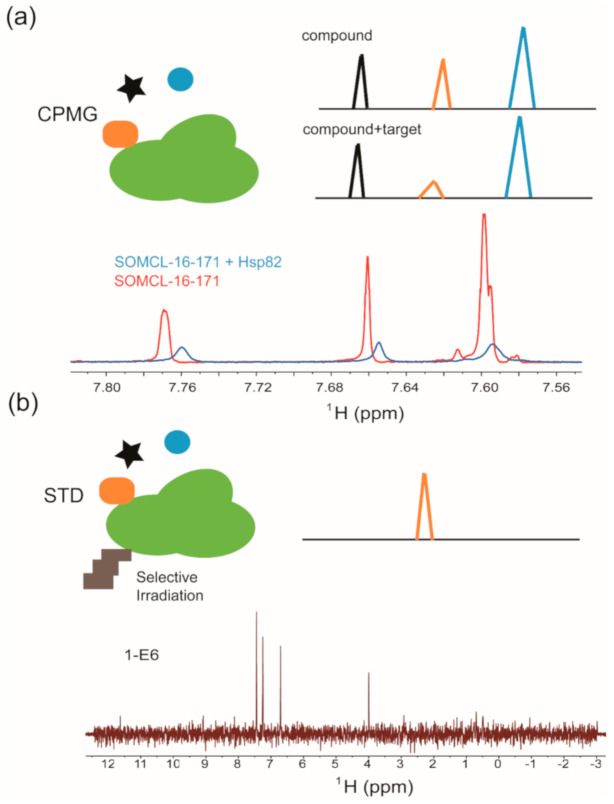
Ligand-observed NMR in hit compound screening and/or validation: (**a**) Cartoon illustration of the Carr-Purcell-Meiboom-Gill pulse sequence (CPMG) experiment. Validation of Hsp82:SOMCL-16-171 interaction by using the CPMG technique is shown as an example. (**b**) Cartoon illustration of the difference spectrum in the saturation transfer difference (STD) experiment. Validation of Hsp90:1-E6 interaction by using the STD technique is shown as an example.

**Figure 3 molecules-26-00576-f003:**
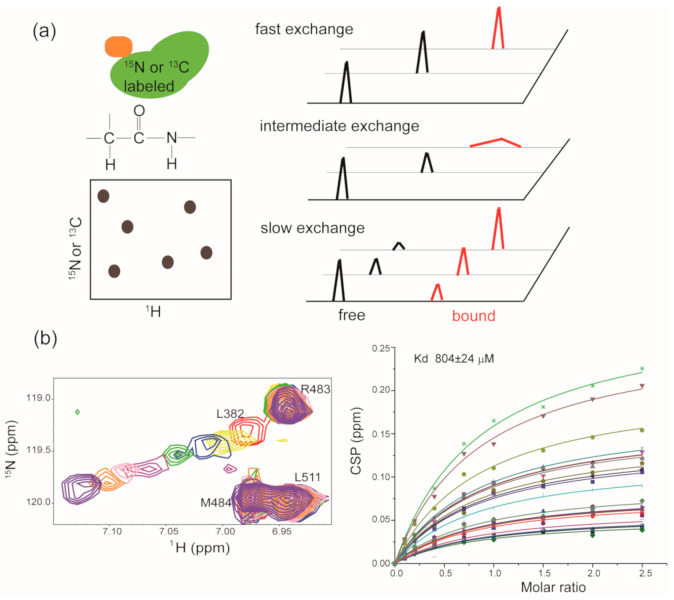
Target-observed NMR in hit compound screening and/or validation: (**a**) Cartoon illustration of the HSQC/HMQC experiments, and each contour in the spectrum serves as a residue-specific fingerprint. The right panel shows the NMR behavior of perturbed residues in target protein upon the addition of weak binder, moderate binder and strong binder, respectively. (**b**) Zoomed view of the superposition of [^1^H, ^15^N] HSQC spectra of Hsp90M upon the titration of SOMCL-16-175. The spectra are colored according to the molar ratio of Hsp90M to SOMCL-16-175 applied in spectrum acquisition: 1:0 (red), 1:1 (yellow), 1:2 (blue), 1:4 (green), 1:7 (magenta), 1:10 (pink), 1:15 (orange), 1:25 (purple). The dissociation constant for the binding of SOMCL-16-175 to Hsp90M was determined by the global fitting analysis of CSP data.

**Figure 4 molecules-26-00576-f004:**
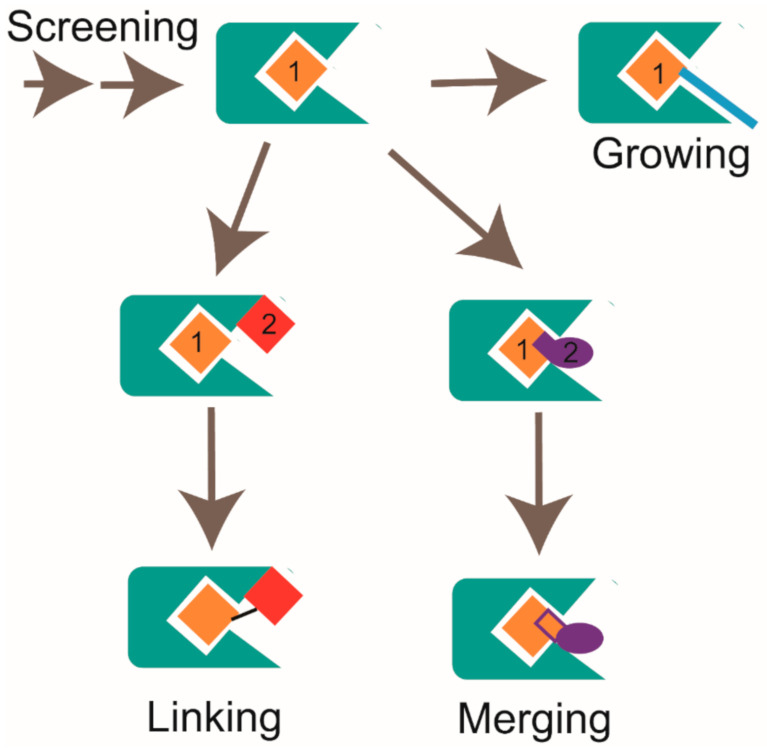
Strategies for fragment optimization, which includes growing, linking, and merging, are shown.

**Figure 5 molecules-26-00576-f005:**
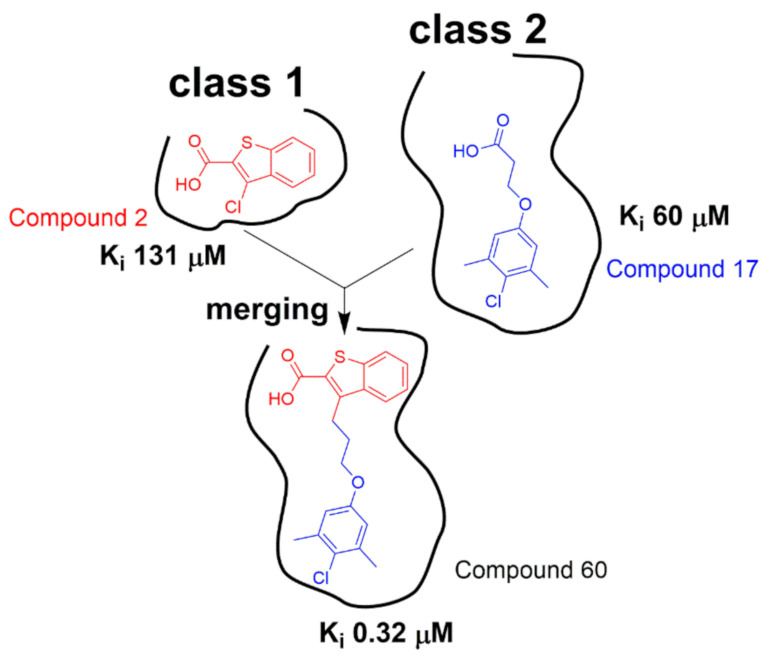
Class 1 hits and class 2 hits were identified to bind to two different but adjacent pockets in Mcl1. Then, linking and merging strategies were jointly applied to the structure optimization of hit compounds (adapted from reference [124]).

**Figure 6 molecules-26-00576-f006:**
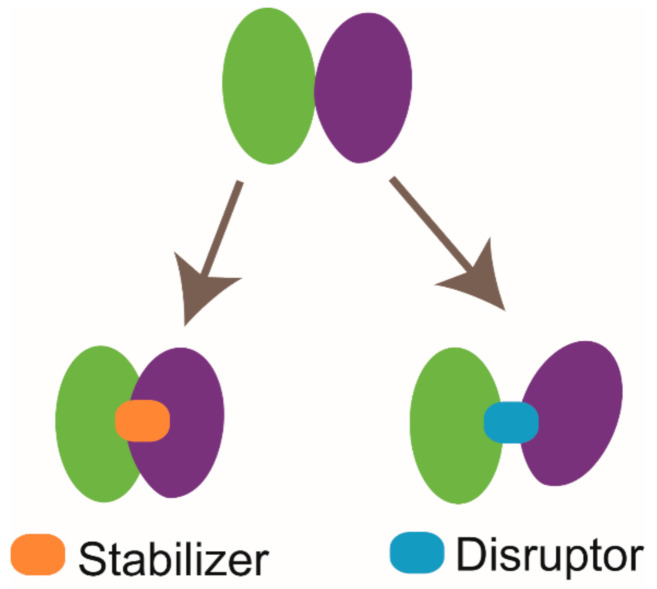
Cartoon illustration of two types of protein-protein interaction (PPI) modulators. The stabilizer functions by enhancing protein-protein interaction, and the disruptor acts by breaking protein-protein interaction.

**Figure 7 molecules-26-00576-f007:**
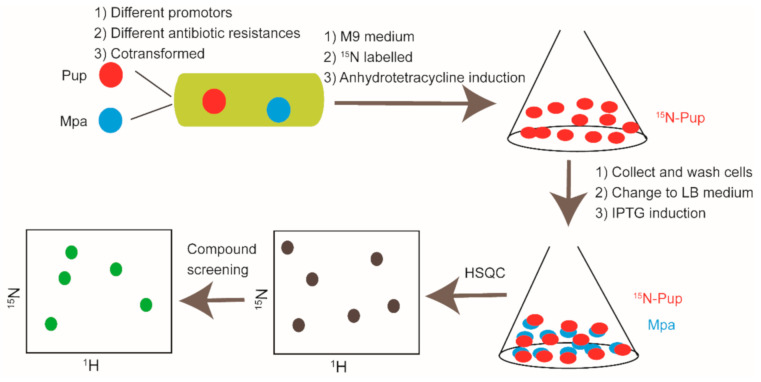
Cartoon illustration of Pup and Mpa PPI modulators discovery by in-cell NMR.

**Table 1 molecules-26-00576-t001:** Advantages and disadvantages of some techniques applied in fragment-based drug discovery (FBDD).

Techniques	Advantages	Disadvantages
Thermal Shift Assay (TSA)	Low cost and high throughput; Identification of hit compounds which modify the target’s thermal stability [68].	False positive results are possible.
Surface Plasmon Resonance (SPR)	High throughput; Detection of binding interactions in real time and in a label-free manner [79].	Targets need to be immobilized; False positive results are possible.
Weak Affinity Chromatography-Mass Spectroscopy (WAC-MS)	High sensitivity; Label free; High efficiency [80].	False positive results are possible [81,82].
X-ray Crystallography	Capable of providing high-resolution and detailed structural information for binding interactions	High quality target crystals are needed.
MicroScale Thermophoresis (MST)	Low cost; high throughput and high efficiency [69].	Fluorescent labeling is needed; False positive results are possible [69].
Ligand-observed NMR	No molecular weight limitation for targets; Capable of detecting weak binders.	False positive results are possible due to compound aggregation.
Target-observed NMR	Capable of providing binding site information; Capable of detecting weak binders.	Isotope labeling is needed; Low throughput; Molecular weight limitation for targets.

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
