# Peer review of "Applications of Solution NMR in Drug Discovery"

_molecules, 2021, doi:10.3390/molecules26030576_

Round 1

Reviewer 1 Report

The paper entitled as “Applications of Solution NMR in Drug Discovery” by Shi et al summarized the progress of NMR applications in drug discovery (fragment-based drug discovery, PPI modulators development, and in-cell NMR for drug discovery), and multiple reported cases were introduced to help readers easily understand the NMR technique used in drug discovery. After reading this paper, researchers in the field of drug discovery will benefit lots from it. Although the authors conducted detailed and systematic work in this paper, significant improvements need to be made before acceptance for publication. Thus, I suggest that this paper can be accepted after major revision. The issues are illustrated as follows:

  1. There are many grammar mistakes all around the paper. Such as, part of the abstract, in the sentence “And NMR has broad applications in different stages of the FBDD process, which including fragment library construction, hit generation and validation, hit-to-lead optimization and working mechanism elucidation etc.”, the word “and” should be deleted and “which including” can be corrected as “which includes” or “including”. A professional English editor is strongly recommended to check/correct grammar and language.
  2. The font of words in the last sentence of page 4 was not consistent with the others. “Normally, the compounds dissolved in DMSO are suggested to be stored in freezer (-20 ℃), and the frequency for freeze-thaw cycles should better be controlled.”
  3. In section 2.3 “Hit-lead optimization”, a specific calculation formular for the ligand efficiency (LE) needs to be provided to explain why LE ≥ 0.3 is considered as a suitable starting point for hit-to-lead optimization.
  4. In section 2.3 “Hit-lead optimization”, when introducing the work of Anders Friberg et al., a figure that describing the optimization process was needed to make it more easier to understand how merging and linking strategies jointly applied to generate new chemicals.
  5. The working mechanism of In-Cell NMR will be illustrated more clearly if some figures (pictures) are added to describe the listed examples.
  6. A table should be added in this paper to illustrate the advantages and disadvantages of NMR and other techniques during different stages of drug discovery.
  7. References part should be check again carefully, should be in accordance with the requirements of the magazine. For example, No15 uses the abbreviation for the journal (J Med Chem 2016, 59, (18), 8189-206.), while No40 uses the full journal name (Journal of Medicinal Chemistry, 2015, 58, (14), 5674-5683.). Similar issues can be found all around the paper. Please check carefully.

Author Response

Dear Review:

We have uploaded our point-by-point responses to you, please see the attachment.

Thank you very much!

Reviewer 2 Report

Fragment-based drug discovery has successfully demonstrated its high capacity and uniqueness in numbers of targets, including the challenging PPIs. NMR has been extensively deployed at different stages of FBDD. The authors well summarized the applications of solution NMR techniques in drug discovery. This manuscript is publishable with minor concerns listed below.

  1. Figure 1 & section 2.4, pseudocontact shifts and associated literatures should also be included. PCSs provides both distance and orientation restraints for elucidation of ligand binding modes.
  2. Figure 3a is a little misleading. It would be better to include C-H bond, which is correlated with the 1H-13C HSQC/HMQC spectra.
  3. Figure 3b & L207. Residues included in the dose-dependent CSP analysis should follow some criterion, as these with small CSPs may suffer from large errors.
  4. Section 3 (L307+), figure 5 is not referred in this section. Maybe I missed it.
  5. L399, “a very difficult if impossible task”?

Author Response

(The authors gave the same response as above.)

Reviewer 3 Report

The authors have written an easily readable review about using NMR spectroscopy in drug design with main focus paid to fragment-based drug design (FBDD). The review is well written with great knowledge of the topic and impresive list of literature references. The text is supported by simple schemes that help to clarify things, however, I would prefer more figures and schemes, especially when the authors describe the case studies. The typical example is Mcl1 binding study, where it is quite difficult for a non-expert to understand the whole topic. Nevertheless, I consider the manuscript clear and readable. The language is on high level and the text is without grammar and  with only several typographical errors, the list of which follows:

  1. 32: inherent – inherently
  2. 143: would – will
  3. 152: which should be removed
  4. 154: …to kinetic off-rate constant… - …to off-rate kinetic constant…
  5. 162: …the false positive introduced… - ….the false positive results introduced….

l.225: What is the meaning of ligand efficiency (LE)? It should be explained here.

  1. 279: nucleus – nuclei
  2. 311: By a combination use of fragment…. – By a combination of fragment….
  3. 312: could – can
  4. 313: would be – will be
  5. 318: selective labeled – selectively labeled
  6. 327: …which was selective.. - …which was more selective…
  7. 330: literatures – literature

Author Response

(The authors gave the same response as above.)

Reviewer 4 Report

The review by Shi and Zhang is a fairly comprehensive coverage of NMR in drug discovery, with a particular focus on fragment screening.  Both ligand-observed and protein-aobserved NMR approaches are covered.  Whereas as a lot of literature was topically covered there were some places where a more thorough description of the literature could be provided.  In addition, given the increased use of both ligand-observed and protein-observed 19F NMR it was surprising this was not covered in more detail, in particular the protein-observed 19F NMR section.  Research by Norton, Prosser, Pomerantz and Wuethrich would be good to cover as there is a substantial body of work.  In a few cases, there does not seem to be appropriate citation.  This was not rigorously checked for all citations but the authors are encouraged to make sure their referencing is accurate.  Finally, the authors are encouraged to get assistance with cleaning up the grammar in the report.  These points should be addressed prior to consideration for publication.

  1. On line 26, NMR has been described to play an important role in drug discovery. The pioneering work was done by Fesik and co-workers in 1996. Their SAR by NMR paper should be cited.

  1. On line 34, MW under 300 is only a guideline put forth by Astex. There is still a debate surrounding what constitutes a fragment.  I would suggest adding a caveat such as “typically  not” exceeding 300 daltons.

  1. Line 45, it is unclear what the authors mean by single target oriented drug discovery. Aren’t typical PPI screens also carried out with a single protein by NMR.  Exceptions are given later on in this report, but its unclear if that is what is meant here.

  1. line 50. I’m not sure targeting PPIs leads to decreased of-target side effects.  Please provide a suitable reference here.

  1. Ref 44 on line 69 is a little dated at this point. There have been a variety of significant advances since then.   Citing more recent work from Pielak, Congang Li, or Selenko may  help here.

  1. On line 98, there has been some research into the average size of a fragment library. Perhaps the authors could provide some harder numbers instead of hundred to thousands.  I thought the average screening library size was somewhere around 1500.

  1. 103. Using “bad actors” seems more like slang.  PAINs compounds are the more accepted term.

  1. Line 124 Fluorine-19 would be the acceptable way to write it here or 19F. 

  1. Line 126. While reference 66 is good to cite, that paper deals with protein-observed 19F NMR.  However it is not covered anywhere in this review and should be updated.  Additional researchers contributing to this field are mentioned above.

  1. line 128. For pooling fluorinated fragments.  Typically CF and CF3 groups are clustered separately due to their differential response, and the large spectral window.  Recently new pulses are available to cover a broader spectral range (see work by  Dalvit), so these aspects are worth citing.

  1. Line 140. The authors attribute CPMG sequences to identify hits based on shifts and broadening.  It was my understanding that this was not typically the case as the molecules are typically in vast excess of protein, so that free ligand resonances mostly dominates and so little shift occurs (although not always the case).  Moreover this is a T2 filtered experiment, which filters out the bound protein resonance contribution, so the data typically  manifests in a decreased in signal intensity, and is less affected by broadening.  The authors should clarify this point.

  1. On line 151, the authors note selective irradiation of methyl resonances, I would suggest adding this is typically done upfield of 0 ppm to avoid small molecule resonances.

  1. Line 152, I believe they should be transfer NOEs rather than intermolecular

  1. On line 164, the authors mention using various ligand observed techniques, but don’t mention what was done or learned. This paragraph should be elaborated.  Ref. 63 seems to be incorrect as this an HSQC NMR screen against BRD4.

  1. Figure 2 is an example where there looks to be experimental data from a paper, copyright permission is needed. Also in the STD experimental data, the spectrum should be  explained.  Is this the difference spectrum?

  1. On line 175, target-observed NMR is described but seems to leave out 1H-13C NMR and 19F NMR.

  1. line 178, what is the MW limitation?

On  line 183  references 78 and 79 don’t appear to be target-based experiments.

  1. On lines 183-188 the examples given are poorly explained. Its unclear how NMR helped to generate the SAR which appears to be the point of this paragraph.

  1. In the discussion of target-based NMR, the aspcets of fragment screening mixtures was not discussed and how that compares to ligand observed.  Relatedly the challenges with deconvolution of the mixtures was not discussed.

  1. Line 205. I would note these ae the total protein and ligand concentrations and not the equilibrium concentrations.

  1. Line 218 The authors note difficulties in deteiming the affinity o binders of moderate affinity.  However see the recent report by Pielak in J. Biomolecular NMR which addresses this point this year.

  1. Line 225. Ligand Efficiency should be defined and noted how it is determined.  It cannot be acquired in all NMR experiments. 

  1. Line 228 what is an organic chemistry approach relative to a med chem approach.

  1. The authors note the time consuming nature of structure determination, but should cite some of the recent work of Julien Orts for addressing this gap.

  1. On line 255, the authors should cite the starting affinity range of the fragments prior to linking to appreciate the affinity enhancement.

  1. On lines 318-319, a reference is needed here.

  1. On line 326, it might be worth noting fragment linking was done in this case.

  1. Line 345, such as what 2D NMR experiment?

  1. On line 351, frog oocytes have also been used and should be cited

Minor: 

1) Line 3, is there another author?

2) Line 39,  I believe you mean X-ray  crystallography

3) Line 41 I don’t believe you mean atomic mechanism.  Perhaps atomic level details

  1. the term “Famous” vendors is not needed.
  2. line 104 Michael receptor should be acceptor
  3. Line 190 footprints should be fingerprints
  4. Medium exchange should be changed to intermediate exchange in the main text and figures.

Author Response

(The authors gave the same response as above.)

Round 2

Reviewer 1 Report

This revised version could be accepted for publication.

Author Response

Dear Reviewer:
Thank you very much!

Reviewer 4 Report

The authors have significantly  revised their manuscript and included additional references to improve their scholarship.  I only have several minor suggestions at this point.

  1. On line 112, the authors attribute low false positives to NMR, however this Is generally  attributed to target-based methods, as ligand-obsered methods can still suffer from significant false positives, many of which are driven by compound aggregation. 
  2. In Table 1, what are the false positives by MS. Defining what type of MS method it is would help clarify this.  If its covalent binding its hard to imagine what a false positive would look like.  If its from WAC, perhaps that’s understandable, but should be referenced.
  3. Also in Table 1, is ligand-observed really a low throughput method compared to other biophysical fragment methods like SPR? Given the fact that larger mixture sizes can be screened with the need for deconvolution, low throughput, seems like an underestimate.
  4. In line 152, the authors still refer to binding events being detected by resonance broadening. Although this may happen in some cases, I believe the main mechanism for CPMG is for filtering out the bound resonance based on a T2 filter, and observation of the ligand free resonance resulting in a drop in resonance intensity  with minimal broadening effects.  The original report by  Hajduk helps address this, https://pubs.acs.org/doi/pdf/10.1021/ja9715962
  5. Finally for the in-cell NMR, Danio rerio, zebrafish embryo’s are an up and coming modality, and could be referenced here.

Minor grammar:

  1. There are also still a few spots to clear up grammatically, a few examples are provided below. 
  2. On line 56, attentions should be attention. 
  3. On line 107, “too crafty”is too colloquial. Perhaps “the promiscuous behavior is non-obvious”
  4. On line 126, it should read the “second most sensitive”

Author Response

Dear Reviewer:
We have uploaded our point-by-point responses to you, please see the attachment.
Thank you very much!
